# Human Activity Recognition Using Cell Phone-Based Accelerometer and Convolutional Neural Network

Ashwani Prasad [1,*], Amit Kumar Tyagi [1,2], Maha M. Althobaiti [3], Ahmed Almulihi [3], Romany F. Mansour [4] and Ayman M. Mahmoud [4]

1 School of Computer Science and Engineering, Vellore Institute of Technology, Chennai 600127, Tamil Nadu, India; amitkrtyagi025@gmail.com
2 Centre for Advanced Data Science, Vellore Institute of Technology, Chennai 600127, Tamil Nadu, India
3 Department of Computer Science, College of Computing and Information technology, Taif University, P.O. Box 11099, Taif 21944, Saudi Arabia; Maha_m@tu.edu.sa (M.M.A.); a.almulihi@tu.edu.sa (A.A.)
4 Department of Mathematics, Faculty of Science, New Valley University, El-Kharga 72511, Egypt; romanyf@sci.nvu.edu.eg (R.F.M.); ayman27@sci.nvu.edu.eg (A.M.M.)
* Correspondence: ashwani.prasad2019@vitstudent.ac.in

**Abstract:** Human Activity Recognition (HAR) has become an active field of research in the computer vision community. Recognizing the basic activities of human beings with the help of computers and mobile sensors can be beneficial for numerous real-life applications. The main objective of this paper is to recognize six basic human activities, viz., jogging, sitting, standing, walking and whether a person is going upstairs or downstairs. This paper focuses on predicting the activities using a deep learning technique called Convolutional Neural Network (CNN) and the accelerometer present in smartphones. Furthermore, the methodology proposed in this paper focuses on grouping the data in the form of nodes and dividing the nodes into three major layers of the CNN after which the outcome is predicted in the output layer. This work also supports the evaluation of testing and training of the two-dimensional CNN model. Finally, it was observed that the model was able to give a good prediction of the activities with an average accuracy of 89.67%. Considering that the dataset used in this research work was built with the aid of smartphones, coming up with an efficient model for such datasets and some futuristic ideas pose open challenges in the research community.

**Keywords:** human activity recognition; sensors; accelerometer; cell phones; dataset; deep learning; convolutional neural network; activity prediction

## 1. Introduction

The accurate measurement of daily activities performed by people has gathered attention for both researchers and the gadget industries. This has given rise to the subject matter of Human Activity Recognition (HAR). The goal of this research is to recognize or predict the action performed by a human subject using certain specialized sensors capable of recording related data [1]. A wide range of activities encompasses HAR including walking, running, sitting, standing, jogging, sleeping, nodding, raising the hand, etc. Wearable devices, smart bands, cell phones and smartphones, are handy pieces of equipment to identify and analyze what a person is doing. Such gadgets provide a wide spectrum of sensors that can be used with ease in day-to-day life with stellar performance and high accuracy. Since people are becoming more conscious about their health, exercise tracking and sleep tracking has become a fashionable trend for many health enthusiasts, who rely heavily on such devices and sensors for such purposes. The behavioral information collected from these sensors is used by researchers to meet needs in the medical and healthcare industries and smart homes [2]. The data recorded in such a manner are also advantageous for many industries and technology giants as it helps them in directing their research and development towards a product that can be potentially launched in the

market in future. Additionally, numerous challenging applications, including automated visual surveillance systems and human–computer interaction (HCI) based on several senses (multimodal), require some level of HAR. Thus, in such cases, an accomplished recognition of human behaviour becomes critical [3].

The research in the field of HAR has been going progressively since the 1980s. One of the reasons is the advent of high-tech computers and smartphones. Smartphones are frequently used by most of the population in most countries of the world. The need to replace human beings with computers to supervise activities of other human beings has kept the research of HAR going. Sensing technologies are widely used in Ambient Assisted Living (AAL) and smart homes. Some existing solutions include physiological, environmental, and vision sensors that frequently assist in human behaviour and activity recognition for health monitoring purposes. Activity recognition is one of the core features of a smart home. Aged and mentally retarded persons suffer from numerous kinds of health issues. As a result, it can lead to improper performance of activities of daily living in such populations. Therefore, the detection of abnormal behaviours becomes indispensable for elderly and dependent people to ensure that they perform their activities correctly with minimum deviation [4]. This can ensure that they are safe and feeling well [5]. Furthermore, suitable types of sensors are also used for different applications such as Falls, Indoor localization, ADL recognition and Anomaly detection. Wearable sensors such as certain accelerometers are used as sensing modalities for all such applications except for Indoor localization [2]. This supports the fact that an accelerometer is a widely used sensor for activity recognition.

There are two major challenges to be tackled for human activity recognition. Firstly, it is tough to manage the huge amount of information produced by the devices and sensors. Secondly, understanding the mapping of data to well-defined movements of the body becomes difficult. Nonetheless, there are a few methodologies that have achieved incredible results in bringing out useful information from the readings produced by sensors [6,7]. Mostly, their experiments required the participants to carry devices in a particular position and orientation. It also required attaching the device to different areas of the body, such as the arms, waist, or wrist. Consequently, the data gathered in such a constrained situation with limitations on device orientation and the number of activities could influence the effectiveness of those models. The ideal scenario is implausible as the device orientation can change with every individual as per their convenience and lifestyle. Depending on their clothing, body shape and size and their behaviour, the positioning of the device is likely to change. Last but not least, several other commonplace activities, namely eating, drinking, talking, typing, writing, smoking, etc., have also been included for recognition. However, their recognition is less reliable because carrying them out requires frequent hand movements. A solution to this problem is to use motion sensors that can be worn on wrists. Other activities, including ascending stairs and descending stairs, cycling and biking are more recognizable when sensors are placed near the pockets. Activities such as these provide better repeating patterns in readings, provided that the sensors are worn at the pocket position. However, the wrist and pocket positions have been primarily used for individuals in seclusion [6,8–11]. For these reasons, HAR models based on artificial intelligence (AI) show a high amount of dependency on the positioning and orientation of the devices and wearable sensors. Additionally, the results of such models also depend indirectly on the type of human activity being carried out.

To address the aforementioned issues, the data used in this work were collected through controlled laboratory conditions. The accelerometer readings were collected from the sensors worn by the subjects as they carried out the six activities—jogging, walking, going upstairs, going downstairs, standing and sitting. The data so obtained are in a raw time-series format. The examples or samples are further accumulated into six classes, where each class is provided with a unique label associated with the corresponding activity. Furthermore, activity recognition is carried out using a predictive model as shall be presented in this paper. The model is based on a Convolutional Neural Network (CNN),

which is an important classification tool in deep learning. Hence, the major contributions of this paper are listed as follows:

1.  Demonstrating the methodology behind the transformation of accelerometer data into appropriate classes, so that it can be fed into the convolutional neural network to perform HAR.
2.  Illustrating that it is plausible to perform HAR with prevalent sensors such as accelerometers, and devices such as commonly used smartphones, and still achieve accurate results. It does not have to include a sophisticated machinery and devices or highly complex algorithms for most of the time to obtain appreciable and useful outcomes. Thus, this paper gives a fundamental and preliminary approach towards HAR.
3.  To validate the proposed 2D CNN model through extensive experiments.

The remainder of this research paper is organized as follows: Section 2 presents the literature review and previous works related to human activity recognition. This section also compares other datasets and techniques that have been used in the past. Section 3 presents the problem description and proposed method and highlights the software tools used. Section 4 illustrates the structure of the dataset, accelerometer data of each activity, data collection, data balancing, data standardization and frame preparation and data splitting. In addition, this section also discusses the development of the proposed CNN model. Section 5 discusses the experimental results along with a comparison between the performance of LSTM and the CNN model used in this paper for HAR. Finally, Section 6 presents the conclusion of this research work and gives ideas for future work.

## 2. Related Work

The research in the field of HAR has been going on for more than a decade. One of the reasons is that most mobile phones today have sensors included in them. The accelerometer has been one of the most widely used mobile sensors for HAR. Moreover, some of the earlier research works used more than one accelerometer located on several areas of the subject's body [12]. It was primarily because mobile phones with accelerometers were less prevalent a decade before. Bao and Intile [13] used five biaxial accelerometers in their research. The experiment consisted of 20 persons. To collect data from them, the accelerometers were placed on distinct parts of the body such as the non-dominant and upper arm, dominant ankle and wrist and the subject's right hip. They recognized 20 activities of daily living by creating models using decision models, instance-based learning, C4.5 and Naïve Bayes classification algorithms. Their experiment deduced that the most suitable position of the accelerometer to place was on the user's thigh. This finding supports the decision implemented during data collection that the pants pocket has been a suitable position for the subject to carry their phones in. On contrary, Krishnan et al. [14] criticized the use of accelerometers on thighs as the collected data were inadequate for classifying activities such as walking, sitting, lying down, etc. Therefore, they emphasized the necessity of multiple accelerometers. Although these systems using multiple sensors can identify varied human activities, they lack practicality and convenience as it requires the users to wear several sensors on their bodies.

Studies on HAR have also been conducted using a combination of sensors of diverse types. Choudhary et al. [15] employed a multimodal sensor apparatus consisting of seven types of sensors, viz., visible/IR light sensor, tri-axial accelerometer, humidity/temperature reader, barometer, microphone, visible light phototransistor, and a compass. They were able to identify activities such as brushing teeth, ascending and descending stairs, standing, sitting, walking and an elevator moving up and down. Such a multi-sensor approach indicates that there lies a remarkable capability in mobile sensor data, especially when a wide range of sensors are incorporated. More recent works based on multiple sensors have been implemented in [16,17]. In these studies, more advanced sensors such as gyroscope, GPS (Global Positioning System), magnetometer, etc., have been used apart from the

accelerometers. Long-themed activities can be recognized more precisely and in a realistic scenario with the aid of such advanced sensors.

In the knowledge area of HAR, other datasets apart from the one used in this paper have been published previously. The UCI (University of California, Irvine) HAR dataset is one of the notable datasets which has been widely used in different works and comparisons. This dataset was proposed in [18] and it was built from the data gathered from inertial sensors embedded in smartphones. There was a total of 30 participants involved in the experiment. Each person performed six activities, viz., walking downstairs, walking upstairs, standing, sitting, laying down and walking. Using the embedded sensors, linear acceleration and angular velocity associated with the three axes were captured with a frequency of 50Hz. The size of the sliding windows was 2.56 s, and the signals were sampled with 50% overlap (128 readings/window) between them. For each record in the dataset, the time and frequency domain variables consisted of a 561-feature vector, which consequently yielded reasonable outcomes. However, the dataset is collected in constrained laboratory circumstances with restrictions on a fixed placement and orientation of the device. Due to such a limitation, in a real-time scenario, the results obtained would not be trustworthy as the users could use their phones in suitable ways.

The next notable dataset is the HARSense dataset, one which has been used in [19]. For robust comprehensive data collection, two sensors (accelerometers and gyroscopes) were used from two different smartphones and two mounting locations in the subject's body were used viz. front pocket and waist. In total, 16 features were collected from the sensors present in both smartphones. This was carried out with the help of an Android application. Some of the prominent features consisted of gravitational acceleration, gravity, rotational rate, and rotational vectors. The dataset generation adopted a physique-based technique in which the data of multiple subjects were clubbed based on the resemblance of their height and weight. The downside of this dataset generation technique is that it is bound to involve participants with a similar physique. This is likely to result in a lesser number of participants, and hence, fewer data.

The Universidade da Coruña (UDC) dataset is yet another promising and flexible dataset used in [1]. The major principle used while collecting data for this dataset is to make it more realistic and applicable in real life which was a lack in dataset developed so far in HAR. The highlighting feature of this dataset is that the smartphone used to collect data can be put in various places of the subject's body and not only in the pockets or around the waist. However, the challenge lies in developing a model that could use such a complex dataset and predict varied states of human activities such as inactive, active, walking and driving.

Furthermore, many other works built their own datasets and carried on with their research on HAR. For instance, in [20], the experiment consisted of nine different orientations of the smartphone. They proposed an online support vector machine (SVM) model to solve the problem. They also made comparisons between their approach with some other state-of-the-art techniques such as Random Forest, CNN, etc. Following the same line, Deep Learning methods have also been used to prepare datasets for HAR. The results in [21] show that these methods are futuristic as they can predict non-stationary activities such as running, jogging, or walking more efficiently. Additionally, it has been found that SVM provides better results in predicting activities that take place for shorter periods. Moving on, models using the long short-term memory (LSTM) approach are becoming popular for both HAR and abnormal human activity prediction [5]. The main advantage of LSTM techniques is that they do not require feature extraction in advance to proceed to model training. However, they require huge datasets for accurate classification outcomes. In addition, they need excess time for model training and to avoid overfitting (and underfitting), a suitable stop criterion is also required. For instance, in [22] and [23], we can see that the LSTM model yielded particularly reliable results. In fact, Bi-LSTMs (bidirectional LSTMs–an improved version of regular LSTMs) used in [22] have shown

exemplary results as the neural network adopts prospective learning, and thus the results are obtained with accuracy as high as 95%.

Moreover, these studies have also used the sliding window approaches in their methods for performing HAR, as can be seen in [24]. Again, this approach depends on the kinds of human activities taken into consideration, placement of the sensor (accelerometer has been used as a sensor), classification methods used and the window size (in seconds). For example, Hemalatha et al. [25] used Frequent Bit Pattern-based Associative Classification (FBPAC) to identify activities such as walking, sitting/standing, lying down and falling. The classification algorithm used a pattern mining based approach to identify the classes corresponding to human activities. The sensor in this study was placed around the chest of the subject. With a window size of 10, they achieved an in-subject classification accuracy of 92%.

The aforementioned methods and techniques have their own advantages. Some of their limitations include:

1. The dataset used in the proposed methods which showed better results have a smaller number of participants. For producing generic HAR outcomes, a bigger size of data collected from several participants is advantageous.
2. Except for one or two proposed methods, there is a meagre analysis and in-depth study carried out on already existing methods, which may still bring comparable results such as the newly developed methods.
3. Less emphasis might be given to the evaluation of the models so developed for performing HAR.

These points are sources of motivation for this paper. Hence, in this paper, an already existing deep neural network method viz. CNN is used to perform the prediction of human activities. Furthermore, it has also been taken care that the model used in this research work is well evaluated before it is put into use. Finally, the dataset used in this paper is that of Wireless Sensor Data Mining (WISDM) as it has been the most widely used, including its first use in [12] and later in many others, including [26–36]. This shows that the dataset used is authentic. Moreover, the dataset boasts a maximum number of participants comparatively. Table 1 shows a qualitative comparison between various datasets used for HAR.

**Table 1.** A brief and qualitative comparison between some datasets used for HAR.

| | UCI HAR | WISDM (Used in This Work) | UDC HAR | HARSense |
|---|---|---|---|---|
| **Human activities studied** | Walking, Walking upstairs, Walking downstairs, Sitting, Standing, Laying | Walking, Jogging, Upstairs, Downstairs, Sitting, Standing | Inactive, Active, Walking, Driving | Walking, Standing, Upstairs, Downstairs, Running, Sitting |
| **Smartphone orientation** | Fixed | Fixed | Free | Fixed |
| **Smartphone positioning** | Waist | Front pants leg pocket | As per the individual's choice | Front pocket and waist |
| **Sensor frequency** | Fixed | Fixed | Not fixed | Fixed |
| **Dissimilar Individuals** | Yes | Yes | Yes | No |
| **Type(s) of sensors used** | Accelerometer; gyroscope | Accelerometer; gyroscope | Accelerometer; gyroscope; magnetometer; GPS | Accelerometer; gyroscope |
| **Number of subjects involved in the study** | 30 | 36 | 19 | 12 |

## 3. Research Methodology

In this section, Section 3.1 presents a description of our problem associated with the recognition of human activities. Next, Section 3.2 gives a brief introduction about the methodology used in general and then the step-by-step approach followed to perform human activity recognition (HAR). Finally, Section 3.3 presents the software tools used in the implementation of this research.

### 3.1. Problem Description

Human activity recognition plays a key role in studying interactions between human beings. It assists in understanding the interpersonal relations of the person. Intuitively, it is difficult to extract the same without human intervention because it consists of comprehending complex factors such as their psychological state, their physiological state and their personality that makes them unique as a person [37]. Additionally, human activity recognition can support the surveillance systems to identify any criminal activity. Moreover, HAR is a significant function for monitoring the health of patients to ensure their well-being. There is a plethora of activities that humans perform in their daily lives. However, this work focuses on the identification of few basic human activities such as walking, jogging, sitting, standing, going downstairs and going upstairs. Each class of activity is labelled from 0 to 5, which shall be differentiated using the CNN model, as mentioned below in Table 2.

**Table 2.** Human activities and their corresponding labels.

| Human Activity | Label |
|---|---|
| Downstairs | 0 |
| Jogging | 1 |
| Sitting | 2 |
| Standing | 3 |
| Upstairs | 4 |
| Walking | 5 |

### 3.2. CNN for Human Activity Recognition

A convolutional neural network, abbreviated as CNN or ConvNet, is a deep learning method that falls under the category of artificial neural networks (ANN). It is mostly applied to analyze visual imagery in common [38]. CNNs have had pioneering results over the years in an array of fields related to pattern recognition, from voice recognition to image processing and video recognition. Additionally, they also have applications in recommender systems [39], image classification, image segmentation and analysis, natural language processing [40], brain–computer interfaces [41] and financial time-series [42]. The striking feature of CNN is that it reduces the number of parameters in neural networks. This advantage has motivated researchers to build bigger models to work out sophisticated tasks, which was otherwise not achievable with classic artificial neural networks [43].

The CNN architecture consists of three layers viz. an input layer, hidden layers and an output layer. The hidden layers consist of convolutional layers which perform a dot product (Frobenius inner product) of the convolution kernel with the input matrix associated with the layers. The activation function related to this dot product is ReLU. A feature map is generated using the convolution operation. Furthermore, it contributes to the input of the next layer. Pooling layers and fully connected layers are the next type of layers that enter the picture. The pooling layers minimize the dimensions of data through the combination of the outputs of neuron clusters into a single neuron in consecutive layers, respectively. Therefore, the computational power required to process the data is reduced. On the other hand, the fully connected (FC) layer extracts the features from the previous layers along with their corresponding filters and accordingly performs classification. In this type of layer, the Softmax activation function is used to classify inputs. The outcome of this function is the probability value lying between 0 and 1.

The input layer requires that the data be reshaped. In 2D CNN, the kernel moves in two directions and hence the adjective "2D" is used to refer to such CNNs. The input and output data of 2D CNN are 3-dimensional. Therefore, the proposed 2D CNN model is given 3-dimensional input data as the values of three accelerometer readings ($x$, $y$, $z$-axes). The accelerometer readings represent time-series data. It is to be noted that the individual accelerometer data of each axis are 2-dimensional. The first dimension represents the time-steps and the second dimension represent the acceleration for the respective axis.

Furthermore, the CNN networks are prone to overfitting data due to the full connectivity of its layers. Overfitting can be regularized using weight decay, skipped connections or dropout [44] methods. During the development of the 2D CNN model, we have used the dropout method to reduce overfitting. Additionally, the activity data are collected from each person involved in the experiment for each activity to minimize overfitting. After pre-processing of data is complete, each record (associated with each activity) is fed into the neural network. Once reshaping of data takes place, the convolutional layer, pooling layer, and the fully connected layers present in the hidden layer perform the necessary operation on the data with Softmax activation at the final layer. The activities can be classified into six classes which implies the creation of 6 output values, one for each activity. Figure 1 indicates the CNN model development to perform HAR.

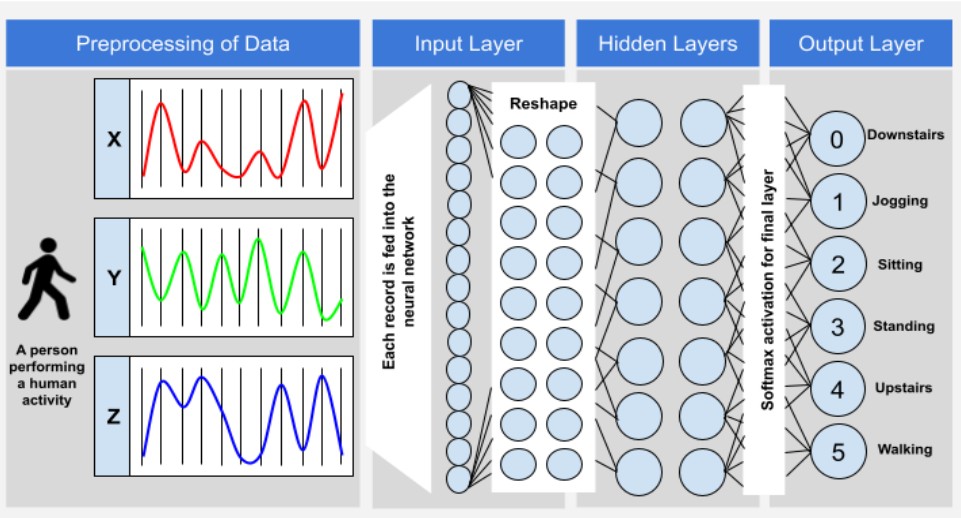

**Figure 1.** CNN model development [45].

Now, we briefly describe the model development process as depicted in Figure 1. The human subject performs any one of the six activities. The accelerometer data are recorded for the three axes (*x*, *y* and *z*). For pre-processing this recorded data, it must be reshaped before feeding it into the neural network. During reshaping, it must be ensured that each person has multiple two-dimensional records which hold 80 time slices for each of the three accelerometer readings. Every record corresponds to a unique label. There are six unique labels, each representing the discussed human activities. Next, those records are fed into the neural network during training. Moving on to the input layer, this layer is a vector with 240 elements (flattened representation of 80-time slices for 3 accelerometer readings each). Furthermore, the three hidden layers have 100 nodes each. The layers are fully connected. There is also an additional layer for reshaping the input into an $80 \times 3$ matrix and a Softmax activation layer as the ending layer. Finally, coming to the output layer, the six available labels, viz., 0, 1, 2, 3, 4 and 5 can be observed in Figure 1. The network would eventually provide the probabilities regarding each output class.

### 3.3. Software Tools Used

The experiment with the CNN model conducted in Section 4 was implemented in the Python programming language. Keras [46] is the major library of Python that was used along with TensorFlow [47]. A Jypyter Notebook was chosen as the implementation platform. Moreover, other important Python libraries used during model development and data analysis were matplotlib [48], scikit-learn [49], pandas [50] and NumPy [51].

Moving on, the model needs to be compiled and fitted. Thus, a mini-batch size is fixed experimentally to 80 samples. This process is supported using an optimizer such as ADAM [52]. It is an algorithm that can be used to rationalize the network weights

iteratively for the concerned training data. It provides better optimization than classical stochastic gradient descent methodology.

## 4. Experiment Study

In this section, the implementation work behind the activity recognition task is explained. Section 4.1 explains the dataset and the data collection process. Section 4.2 describes the six activities concerning the accelerometer readings. Section 4.3 explains how data are balanced in the dataset. Section 4.4 discusses the data standardization and frame preparation for the study. Section 4.5 briefly explains the splitting of data for training and testing purposes. Finally, Section 4.6 presents the development of the 2D CNN model along with model evaluation and the confusion matrix.

### 4.1. Data Collection

This research uses the WISDM dataset [12] as described in Table 1. A total of 1,098,209 examples are present in the dataset. A total of 36 participants participated in the experiment. To collect data for this experiment, every participant was required to carry a smartphone running on the Android operating system during the performance of each of the six activities. The subjects were required to carry their Android phones in a pocket situated in the front portion of their pants. Furthermore, they were advised to sit, walk, stand, jog, climb upstairs and go downstairs for specific durations of time.

An Android application, running on the smartphone, controlled the data collection from the accelerometer during activity performance by the subjects. The application used in the experiment comprised a graphical user interface, which permitted the experiment supervisor to record the subject's name and label the activity that the subject performed. Furthermore, the application also provided a feature to start and stop collecting data. In addition, the application enabled the supervisor to control the type of data from different sensors (e.g., gyroscope, accelerometer) and to set the frequency of data collection. Overall, the data from the accelerometer were collected every 50 milliseconds at the rate of 20 samples per second.

For example, Table 3 shows a part of the dataset with the activity details of participant with user ID 33. The participant is performing the jogging activity. The time column consists of the timestamps in numeric form, which is generally the phone's uptime in nanoseconds. The corresponding coordinate values of the three axes are also included in the table.

**Table 3.** Activity details of a typical subject of the experiment.

|   | User ID | Activity | Time | $x$ | $y$ | $z$ |
|---|---------|----------|------|-----|-----|-----|
| 0 | 33 | Jogging | 49105962326000 | $-0.6946377$ | 12.680544 | 0.50395286 |
| 1 | 33 | Jogging | 49106062271000 | 5.012288 | 11.264028 | 0.95342433 |
| 2 | 33 | Jogging | 49106112167000 | 4.903325 | 10.882658 | $-0.08172209$ |
| 3 | 33 | Jogging | 49106222305000 | $-0.61291564$ | 18.496431 | 3.0237172 |
| 4 | 33 | Jogging | 49106332290000 | $-1.1849703$ | 12.108489 | 7.205164 |

### 4.2. The Activities

In this research study, the activities that are taken into consideration are: walking, going upstairs, going downstairs, sitting, jogging and standing. There are two reasons why these activities were selected. Firstly, these activities are quite common, and many people perform them in their everyday lives. Secondly, these activities also involve movements of the body that take place for a considerable amount of time, hence recognizing them becomes an easy task. Moreover, most of these activities show repetitive patterns in body movements, which can make their recognition easier [12]. When the data are recorded for respective activities, the accelerometer readings (acceleration values) are recorded in the three axes, viz., $x$-axis, $y$-axis and $z$-axis. For this study, the horizontal motion of the subject's leg is captured in the $x$-axis. The upward and downward motion of the body is

captured in the *y*-axis. Lastly, the forward motion of the subject's leg is captured in the *z*-axis.

Figure 2a–f shows six plots for each activity performed by a typical human subject who had participated in the experiment. Before depicting the accelerometer values of the three axes in their graphical forms, a check for null or missing values was performed. It was found that there were no null values in the reading, thus ensuring that the data are well-filtered before representation. Every plot consists of three subplots representing acceleration values concerning the *x*, *y*, and *z*-axis. As mentioned before, the data are recorded at a frequency of 20 Hz, i.e., 20 samples per second. As only 180 records are considered from the beginning, each plot illustrates an interval of 9 s for the respective activities (calculation: $0.05 * 180 = 9$ s). It could be observed that sitting and standing activities as seen in Figure 2e,f, respectively, do not display recurring behaviour but do possess distinguishing patterns with respect to points on the *x*, *y* and *z* axes. In contrast, the rest of the four activities (Figure 2a–d), which involve carrying out repeated motions, do show recurring patterns in the plots. An important aspect of the plots in Figure 2 that should be noted is that the values of the y-axes have the highest magnitudes of acceleration compared with other axes. This is because of Earth's gravity, which causes an acceleration of 9.8 m/s$^2$ towards the centre of the earth. Due to this, the accelerometer measures 9.8 m/s$^2$ in the direction of the Earth's centre.

The approximate periodic patterns for the activities shown in Figure 2a–f show a relationship between time and accelerometer reading values. Additionally, as per expectation, standing and sitting activities have constant acceleration values when there is a negligible motion before and after sitting or standing. The plot for walking shown in Figure 2a shows more variations and many peaks as it involves more motion. The distance between two consecutive peaks on the *y*-axis and *z*-axis represents the time taken by the subject to perform a stride while walking. The adjacent values of the *x*-axis plot are lesser in magnitude comparatively and display the peaks in association with the remaining two axes. For jogging activity (Figure 2d), similar trends such as walking are seen for the data related to the *y*-axis and *z*-axis. Nonetheless, the extent of magnitudes of acceleration on the *y*-axis is more than that present in walking, with an observable shift in the negative direction. Additionally, it can be observed that jogging involves comparatively more variation and higher frequency due to repetitive motions of the body.

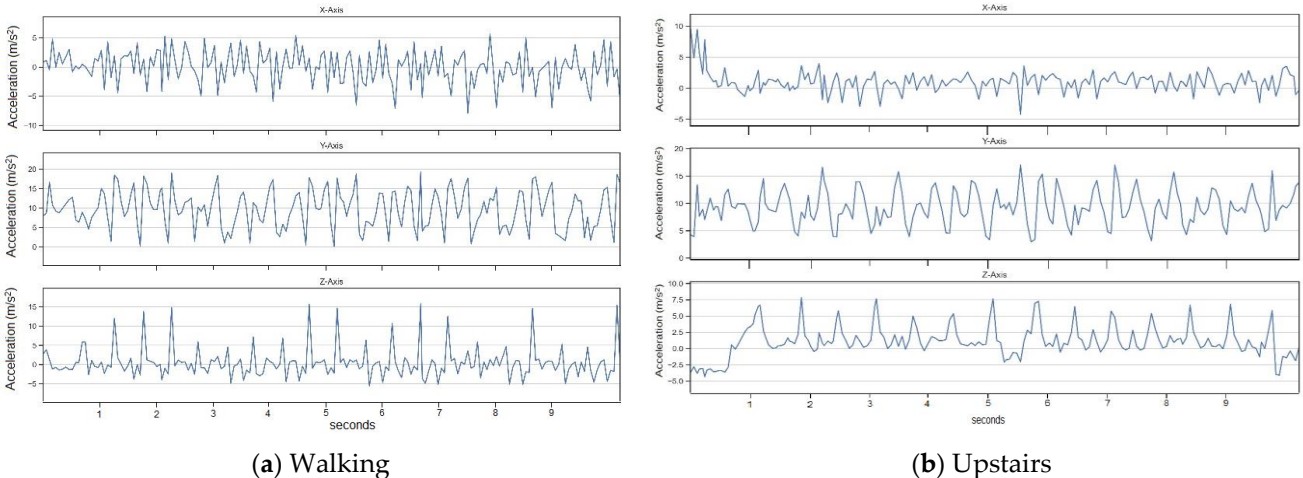

(**a**) Walking  (**b**) Upstairs

**Figure 2.** *Cont.*

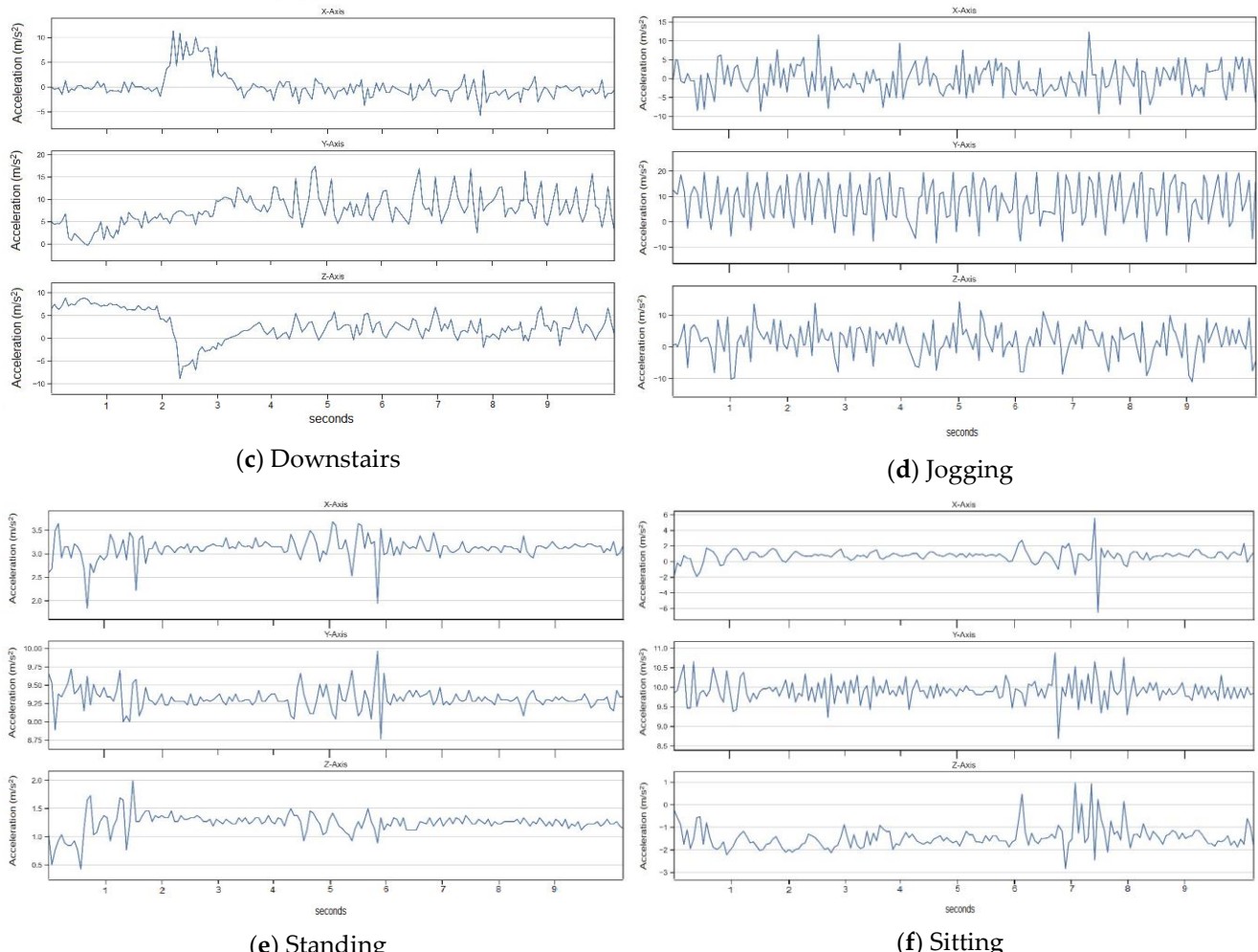

**Figure 2.** Plots representing accelerometer readings for the six human activities (a–f). (**a**) Accelerometer readings for Walking. (**b**) Accelerometer readings for Upstairs. (**c**) Accelerometer readings for Downstairs. (**d**) Accelerometer readings for Jogging. (**e**) Accelerometer readings for Standing. (**f**) Accelerometer readings for Sitting.

For downstairs, as seen in Figure 2c, each small peak in the *y*-axis plot indicates that the subject is going down by one stairstep. Similarly, the *z*-axis values show that the subject is moving down each stair, but with acceleration in the negative direction. Next, the *x*-axis plot displays a progression of small peaks in a semi-regular fashion, wherein the acceleration is wavering between positive and negative magnitudes. For upstairs (Figure 2b), there is a sequence of semi-regular peaks in the *x*-axis with greater magnitudes initially. In the *y*-axis, the peaks are occurring in regular intervals. The *z*-axis data have slightly spaced peaks, which shows that the subject takes a longer time to climb upstairs in comparison with downstairs.

### 4.3. Data Balancing

The initial data from the dataset are highly unbalanced. It can be seen from Figure 3 that a greater amount of data for walking and jogging activities is present compared with other activities. This is because the number of training examples or samples for those two activities have been collected in abundance compared with the rest of the activities during the actual experiment. The training examples or samples are crucial for model development and human activity prediction. As CNNs are supervised deep learning models, they require huge set of input data and known responses to the output data for learning. Greater number of samples assist in better learning of the CNN model and hence lead to better prediction, provided that the number of samples for all the six activities are

equal. It is also clear from Figure 4 that the total number of participants in the experiment is 36.

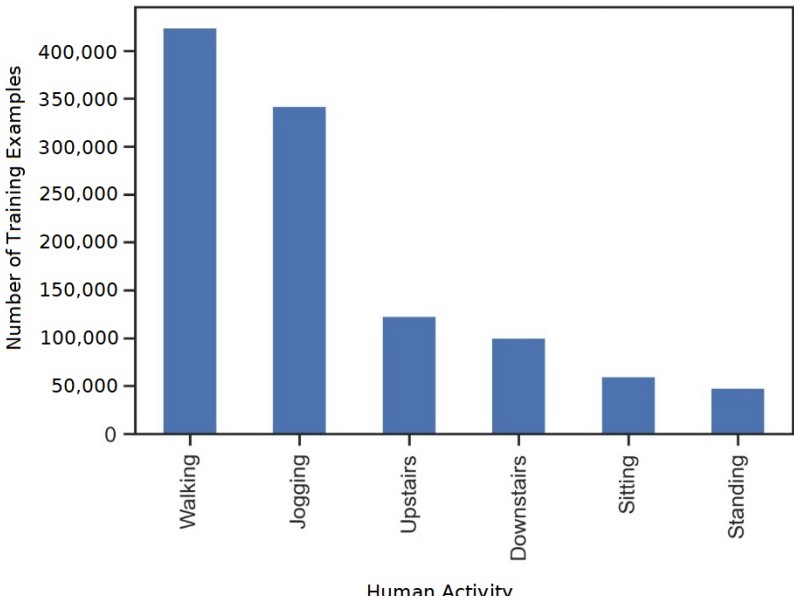

**Figure 3.** Training examples by activity type.

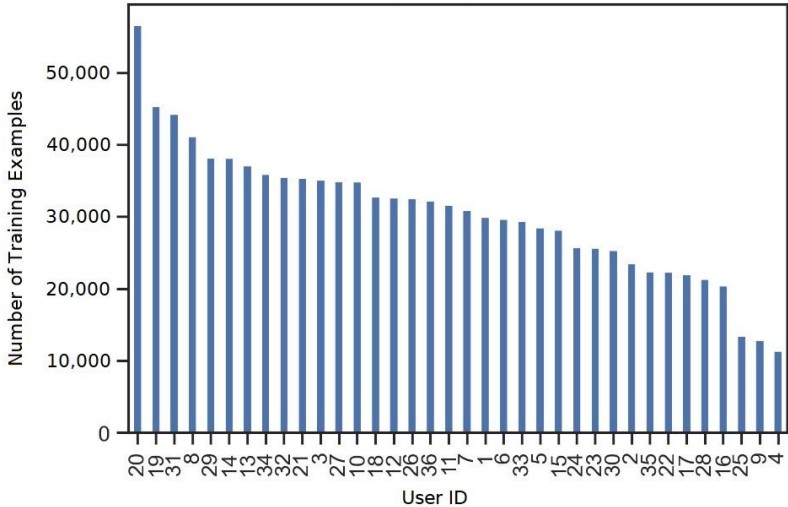

**Figure 4.** Training examples by user.

Furthermore, from Figure 4, it is obvious that user 20 has recorded the highest amount of data for the experiment. On the other hand, user 4 has recorded the least amount of data. Hence, there is non-uniformity in the recording of data by participants as well. However, this non-uniform distribution of training examples is less of a concern as we are more interested in the greater number of examples for different activities with little stress over who performs them. However, for better prediction and recognition of human activities, it is crucial to have balanced data for each of the six activities. From Figure 3, we can observe that the data are highly skewed towards walking and jogging. If this issue is avoided, it might result in overfitting during training, which could lead to an unwanted performance drop of our machine learning model [53]. Since equal importance is ought to be given for all the six activities, it is necessary to observe the activity which has the least number of examples. It is a standing activity with 3555 examples. Therefore, data are balanced by

considering 3555 training examples from each of the six activities. The balanced number of training examples by activity type is shown in Figure 5.

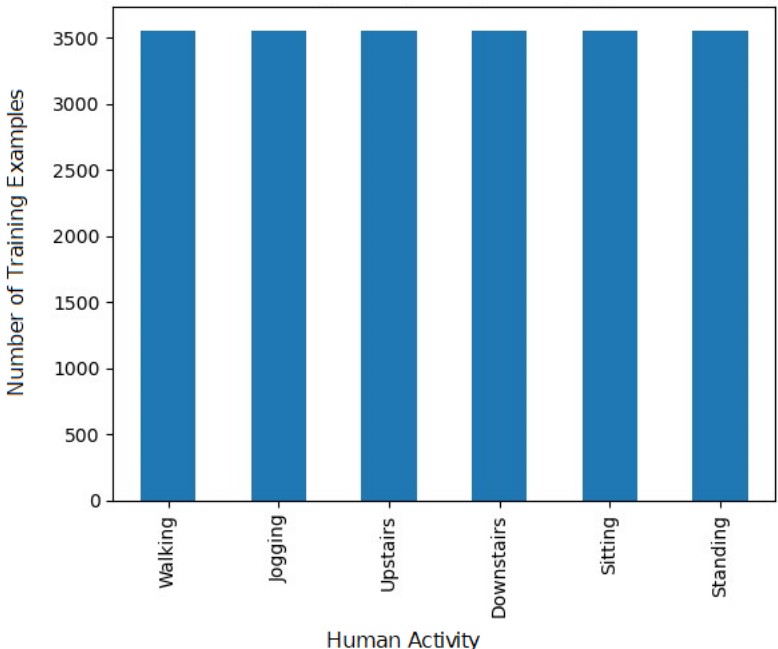

**Figure 5.** Balanced number of training examples for each activity type.

Furthermore, the columns present in Table 3 are in string format. However, the values of $x$, $y$ and $z$ are required in the format of floating-point numbers for further processing. Thus, the datatype of $x$, $y$ and $z$ column is changed to "float64", representing floating-point values. After the dataset is balanced, all the activities are brought in a single Data frame. Moving on, the activity variables are converted into categorical values. Labels (output/target variables) from 0 to 5 are given to activities, viz., going downstairs, jogging, sitting, standing, going upstairs and walking, respectively, with the help of an array of size 6.

### 4.4. Data Standardization and Frame Preparation

The values of $x$, $y$ and $z$ can be seen in Table 3 for the jogging activity performed by user 33. It can be observed that the respective values show a variance of greater magnitude between them in their respective columns. To decrease variance among the data, a standard scaling technique was used, which resulted in values with much lesser variance, as seen in Table 4.

**Table 4.** Standardized data with lesser variance in $x$, $y$ and $z$ values.

|  | $x$ | $y$ | $z$ | **Label** |
|---|---|---|---|---|
| 0 | 0.000503 | 0.000503 | 0.000503 | 5 |
| 1 | 0.073590 | 0.073590 | 0.073590 | 5 |
| 2 | −0.361275 | −0.361275 | −0.361275 | 5 |
| 3 | 1.060258 | 1.060258 | 1.060258 | 5 |
| 4 | −0.237028 | −0.237028 | −0.237028 | 5 |
| . . . | . . . | . . . | . . . | . . . |
| 21325 | −0.470217 | −0.470217 | −0.470217 | 3 |
| 21326 | −0.542658 | −0.542658 | −0.542658 | 3 |
| 21327 | −0.628514 | −0.628514 | −0.628514 | 3 |
| 21328 | −0.781444 | −0.781444 | −0.781444 | 3 |
| 21329 | −0.800225 | −0.800225 | −0.800225 | 3 |

The features are standardized by removing the mean and scaling to unit variance. For instance, consider a sample X. The standard score of the sample X is calculated as:

$$z = (X - u)/s$$

where u is the mean of the training samples and s is the standard deviation of the training samples. During implementation to obtain the standardized data, the data were firstly fit and then transformed using the fit_transform () method under the sklearn.preprocessing. StandardScaler class. These data are further converted into a data frame using the pandas. DataFrame class.

Data frame preparation is necessary as the subjects will be performing their activity for a random period of time, but only a certain amount of data can be fed into the neural network at a time. Therefore, with the help of a Data frame, it is possible to analyze the data in two dimensions. It also helps in dividing the data into batches that need to be processed by the model. For this purpose, the frequency is taken as 20, and the frame size is taken to be equal to 80 (frequency $*$ 4). This implies that 80 $*$ 3 samples will be fed to the model in a batch, considering all three dimensions. The hop size is taken as twice as the frequency, i.e., equal to 40. This implies that the advancement on the Data frame will be made with 40 data samples. Thus, the number of records in each dataset can be calculated as:

(No. of training samples of an activity $\times$ Total no. of activities/hop size) = 3555 $\times$ 6/40 $\approx$ 533.

### 4.5. Splitting Data for Training and Testing

The complete dataset is split into a training set and a test set as it is necessary for training and evaluating the model. While splitting the data, care was taken that the data present in the test set did not affect the training set data. This is ensured by allocating 80% of available data for training and the remaining 20% of data for testing. This approach can also prove beneficial for analyzing the overall performance of the model during training and validation. The idea behind splitting is to let the neural network learn from the data generated by few persons who have been through the experiment. Additionally, since each two-dimensional record should hold 80 time slices for the accelerometer readings, the data are reshaped as follows:

**Before reshaping**

Shape of the training set = (532, 80, 3)

Shape of the testing set = (532)

where 532 represents number of samples which is approximately equal to the calculated number of records, which is 533; 80 is the width, i.e., the number of time slices is 80 and 3 represents the height (3 accelerometer readings from $x$, $y$ and $z$ axes).

**After reshaping**

Shape of training set = (425, 80, 3); since, 532 $\times$ 0.8 $\approx$ 425

Shape of testing set = (107, 80, 3); since, 532 $\times$ 0.2 $\approx$ 107

where 425 and 107 are the number of samples in the training and the testing set, respectively. The reason for reshaping is to ensure that the data input to the model are in the correct shape. Next, we are interested in knowing how well our model predicts the bodily motions of persons it has not come across.

### 4.6. The 2D CNN Model

After the prerequisite steps were completed, the next step was the development of a two-dimensional convolutional neural network. The data were, therefore, made ready to be processed by Keras. Moving on, the model was developed using five layers. The first layer is the sequential layer, which enabled us to add further layers to the neural network. This layer also performs reshaping of the data by bringing them into the "old" format. The

filter size for this layer is set to 16. Following the previous layer, the next three layers are the hidden layers. Six classes were used in the model (going downstairs, jogging, sitting, standing, going upstairs, walking). The aforementioned layers use the Rectified Linear Unit (ReLU) as a universal activation function and have a filter size of 32. The kernel size for the first four layers is (2,2), which implies that the height and width of the convolution window are 2 and 2, respectively. Lastly, the fifth layer comprises the dense layer which identifies the six classes and runs a Softmax activation function for multi-class probability calculation. To reduce overfitting, a dropout of 0.1 is used for the sequential layer and a dropout of 0.2 is used for the next three hidden layers. Next, for the dense layers, a dropout of 0.5 is used. The model is finally built. The input, convolution, pooling, fully connected and output layer is depicted in Figure 6. The next step is the compilation of our model. This is carried out by executing ADAM optimizer with a learning rate of 0.001, the loss is taken as sparse categorical cross-entropy and the metrics are based on accuracy. Furthermore, the training data are fitted into the model with 10 epochs and the training begins. The first epoch had achieved an accuracy of only 24% with training, whereas the validation was 42.06% accurate. Coming to the tenth epoch, the training was highly accurate with 93.88% accuracy whereas the validation was 89.72% accurate. Section 4.6.1 describes how well the model has learnt, and the final confusion matrix for activity prediction is shown in Section 4.6.2.

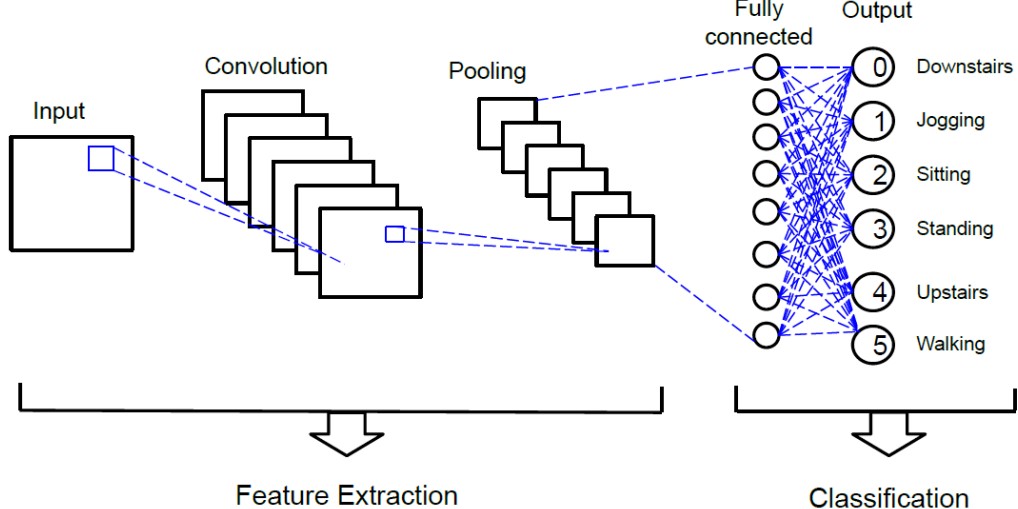

**Figure 6.** Different layers of the 2D Convolutional Neural Network.

4.6.1. Model Evaluation

After the training is finished, it is necessary to gauge our model so that we can be sure that it can predict human activities well enough. For this purpose, two learning curves were plotted, which are presented in this paper. Figure 7 shows the plot of training and validation accuracy values for all the ten epochs. Next, Figure 8 shows the plot of training and validation loss values for all the ten epochs.

From Figures 7 and 8, we can conclude that the CNN model has been trained with high accuracy. Additionally, in Figure 8, the validation loss is less than the training loss. This shows that the developed model is neither overfitting nor underfitting.

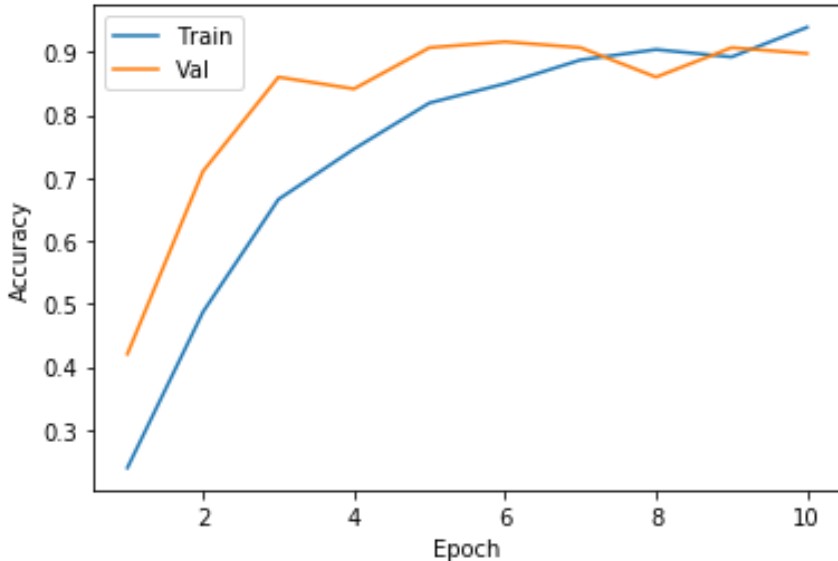

**Figure 7.** Graph depicting model accuracy.

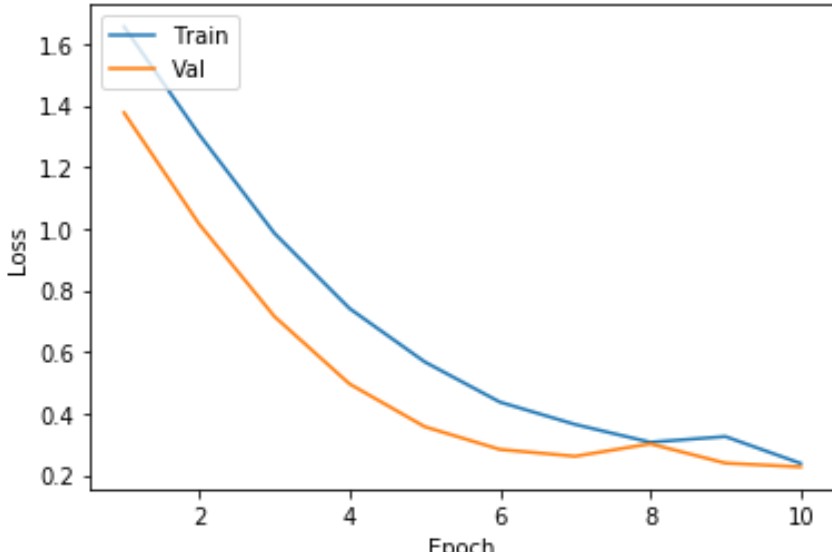

**Figure 8.** Graph detecting model loss.

4.6.2. The Confusion Matrix

Finally, the six activities are predicted using a confusion matrix. For this, the test data are passed in an appropriate function, and as a result a 6 × 6 matrix is obtained with the diagonal squares containing the probabilities of recognition of each of the six classes (See Figure 9).

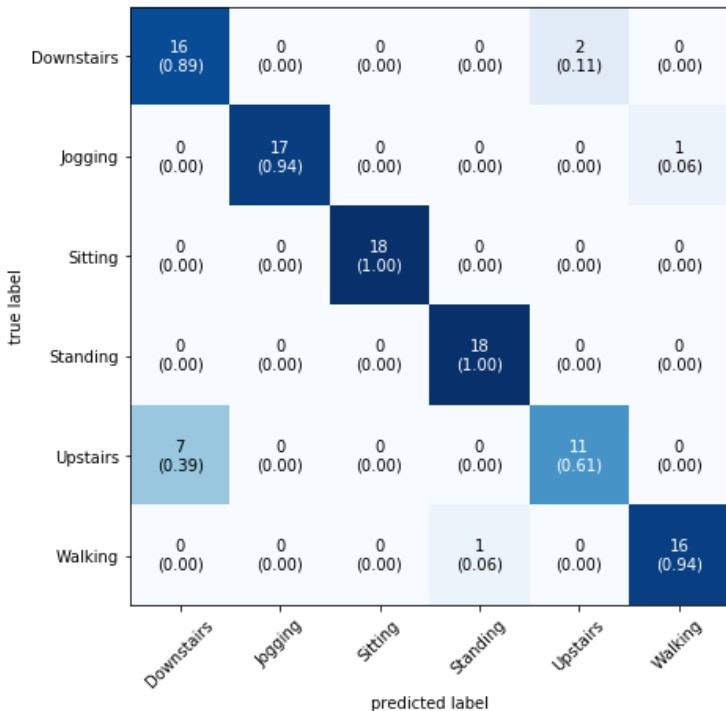

**Figure 9.** Confusion matrix showing the prediction scores of each activity.

## 5. Results and Discussion

Based on the confusion matrix as seen in Figure 9, we conclude that the proposed 2D CNN model was successful in predicting the human activities with the accuracies seen in Table 5.

**Table 5.** Human activity prediction scores.

| Activity | Prediction Accuracy (%) |
|:---:|:---:|
| Downstairs | 89% |
| Jogging | 94% |
| Sitting | 100% |
| Standing | 100% |
| Upstairs | 61% |
| Walking | 94% |

From Table 5, it can be concluded that the 2D CNN model was perfectly able to recognize sitting and standing activity. Moreover, activities such as jogging, walking and downstairs were predicted with good accuracy. However, the upstairs activity has the least accuracy because the model becomes confused with the downstairs activity due to the similar nature of both the activities. The average accuracy of predicting the six activities is 89.67%

Now, to substantiate the efficacy of the method that we put forward in this paper, we compare it with a state-of-the-art method, namely a popular method such as the LSTM. The results are summarized in Table 6.

From Table 6, it can be said that LSTM fares well in the prediction of all the six activities with high accuracy. However, the 2D CNN model performs better in activities such as downstairs activity, standing and sitting. Additionally, the average accuracy of predicting the same activities using LSTM is found out to be approximately 92.33%. Even though LSTM has higher average accuracy, the 2D CNN model still fares well in activity prediction.

**Table 6.** Comparison of accuracy between 2D-CNN and LSTM.

| Activity | LSTM Accuracy (%) | 2D CNN Accuracy (%) |
|---|---|---|
| Downstairs | 86% | 89% |
| Jogging | 96% | 94% |
| Sitting | 92% | 100% |
| Standing | 94% | 100% |
| Upstairs | 87% | 61% |
| Walking | 99% | 94% |

## 6. Conclusions

This paper presented a 2D CNN model for recognizing and predicting simple human activities with the help of an existing dataset based on accelerometer readings. The proposed model identifies and predicts the majority of the six activities with a high degree of accuracy—over 85% of the time, on average. The approach of building the convolutional neural network in a two-dimensional format and its application in HAR is the highlighted feature of our work. This paper also shows how important research on human activity recognition has become in present times, with newer methods coming over time.

In addition, this paper also shows how learning evaluation is necessary before proceeding towards model development. This helps in comparing the learning scores with the actual prediction scores of the model, and thereby a consistency check can be performed. Moreover, the results obtained in this paper were comparable with a renowned state-of-the-art method—the LSTM.

Our future work can focus on working on newly developed datasets that are more aligned with real-life scenarios, consisting of a greater number of activities to predict and a bigger number of participants. Otherwise, an effort can also be made to collect data, per se, with a greater number of participants with different lifestyles, behaviour, ageing, sex, etc. Another future work could include abnormal human behaviour prediction. With the help of a functional activity recognition system, one can supervise the dependent persons such as the elderly people in smart homes and evaluate their activity level for healthcare services. Moreover, for all the residents staying in smart buildings, human activity recognition can be utilized in checking their comfort level concerning factors such as temperature and humidity. Last but not least, 1D and 3D approaches towards building the CNN model could be tried out for predicting human activities in future.

**Author Contributions:** All authors have contributed equally to this paper. Conceptualization, A.P.; Methodology, A.P.; Supervision, A.K.T., M.M.A., A.A., R.F.M. and A.M.M.; Writing—original draft, A.P.; Writing—review & editing, A.P. All authors have read and agreed to the published version of the manuscript.

**Funding:** We deeply acknowledge Taif University for supporting this research through Taif University Researchers Supporting Project Number (TURSP-2020/328), Taif University, Taif, Saudi Arabia.

**Institutional Review Board Statement:** Not applicable.

**Informed Consent Statement:** Not applicable.

**Data Availability Statement:** The WISDM (Wireless Sensor Data Mining) data set used for this research work was taken from https://www.cis.fordham.edu/wisdm/dataset.php.

**Acknowledgments:** This research work gratefully acknowledges the Centre for Advanced Data Science, VIT Chennai for conducting a research internship program in the month of July 2021 (under the supervision of Amit Kumar Tyagi) and supporting the research activity, without which it would not have been possible to complete this research paper. And we deeply acknowledge Taif University for supporting this research through Taif University Researchers Supporting Project Number (TURSP-2020/328), Taif University, Taif, Saudi Arabia.

**Conflicts of Interest:** The authors declare no conflict of interest.

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
