# Peer review of "Human Activity Recognition Using Cell Phone-Based Accelerometer and Convolutional Neural Network"

_applsci, doi:10.3390/app112412099_

Round 1
Reviewer 1 Report
The article is proposing a 2D Convolutional Neural Network for the classification of human activity using the acceleration signal measured with a cell phone. The overall flow of the article is fine, however, there are some methodological ambiguities that should be addressed in the revised manuscript. The authors should address the following comments in the revised manuscript.
- The authors mentioned that “During reshaping, it must be ensured that each person has multiple two-dimensional records which hold 80 time slices for each of the three accelerometer readings.” However, I cannot find any explicit details on the reshaping of data from the accelerometers. Add such detail in the revised version.
- No detailed discussion on the architecture of the network such as the filter size, the arrangement of layers (convolutional, max pooling, drop out etc.). Also, the authors mentioned the issue of overfitting but did nothing to fix it in the current network. Such detail should be added to make the results reproducible and reliable.
- In Table 2, time appears too strange where the authors said that the unit of time is seconds. However, if the time is shown in seconds, then the time is equivalent to almost 157,876 days which does not make sense for a single point measurement. The authors should correctly describe the meaning and unit of time in Table 2.
- The authors mentioned that “For this study, the horizontal motion of the subject’s leg is captured in the x-axis. The upward and downward motion of the body is captured in the y-axis. Lastly, the forward motion of the subject’s leg is captured in the z-axis.” For a better explanation, I recommend adding a schematic of humans with the x, y, and z directions marked on it.
- For figure 2, show the label for the y-axis, and put the numbers and label on the x-axis for getting insights into the data.
- To better understand Figure 3, I recommend explicitly mentioning the meaning of the number of samples.
- In section 4.4, the authors mentioned data standardization without any mathematical details. I recommend showing detail of the standardization process.
- I cannot find specific details on the data provided as input to the CNN. The input data to the network was 1D, 2D, or 3D, explicitly mentioning the types and the preprocessing if the data is 2D or 3D.
- In section 4.5, the authors said, “For our implementation, we have decided to split the data into three dimensions.” It difficult to understand the meaning of 3D splitting. Add more details.
- Figure 6 shows the training and validation process with respect to the number of epochs, however, in the data splitting the authors only mentioned the training and testing data and said that the testing data is not seen by the network during the training process. Explicitly mention the validation data.
- From the plots of training and validation accuracies and losses, the 10 epochs do not seem to be sufficient. I recommend showing the results for 30 epochs to have a full picture of the training process.
- The topic of human activity recognition has been extensively, and some authors have also shared their source codes. I recommend adding a table showing the comparison of the current work with all the previously published work and showing the contribution of the article from such a comparison.
- The average as well per class accuracy of LSTM is better than the proposed approach, I wonder that if LSTM is performing better than 2D CNN, why the authors did not employ LSTM instead of CNN.
- The authors are referring to the employed CNN as 2D CNN, however, it is difficult to fully grasp the idea. If the reason for calling it 2D CNN is that the data provided at the input is 2D, then it should be clearly mentioned in the text. Also, details on how the 1D acceleration signal was transformed to 2D should be added.
Reviewer 2 Report
The paper presents a method undertake human activity recognition using 2D CNN. The paper is interesting and well written. The following comments should be considered:
- Figure 2 : the x-axis is missing and the units are missing from the y-axis
- Figure 3 : units for the y-axis are missing
- Figures 4-7 : units are missing
- Figure 2 : Did the author filter the data? The author should comment on that.
- The author mentions contribution 2 is : “Illustrating that it is plausible to perform HAR with prevalent sensors and vices and still achieve accurate results.” This has already been reported in the literature. Can the author comment on that?
Round 2
Reviewer 1 Report
The authors have addressed all my comments in the revised manuscript which has improved the quality of the paper. I recommend the article for publication in the current form.
Reviewer 2 Report
ok